# Robust Zero-Watermarking of Color Medical Images Using Multi-Channel Gaussian-Hermite Moments and 1D Chebyshev Chaotic Map

**DOI:** 10.3390/s22155612

**Published:** 2022-07-27

**Authors:** Doaa Sami Khafaga, Faten Khalid Karim, Mohamed M. Darwish, Khalid M. Hosny

**Affiliations:** 1Department of Computer Sciences, College of Computer and Information Sciences, Princess Nourah bint Abdulrahman University, P.O. Box 84428, Riyadh 11671, Saudi Arabia; dskhafga@pnu.edu.sa; 2Department of Computer Science, Assiut University, Assiut 71516, Egypt; mohamed_darwish@aun.edu.eg; 3Department of Information Technology, Zagazig University, Zagazig 44519, Egypt; k_hosny@zu.edu.eg

**Keywords:** zero-watermarking, color medical images, Gaussian–Hermite moments, multi-channel

## Abstract

Copyright protection of medical images is a vital goal in the era of smart healthcare systems. In recent telemedicine applications, medical images are sensed using medical imaging devices and transmitted to remote places for screening by physicians and specialists. During their transmission, the medical images could be tampered with by intruders. Traditional watermarking methods embed the information in the host images to protect the copyright of medical images. The embedding destroys the original image and cannot be applied efficiently to images used in medicine that require high integrity. Robust zero-watermarking methods are preferable over other watermarking algorithms in medical image security due to their outstanding performance. Most existing methods are presented based on moments and moment invariants, which have become a prominent method for zero-watermarking due to their favorable image description capabilities and geometric invariance. Although moment-based zero-watermarking can be an effective approach to image copyright protection, several present approaches cannot effectively resist geometric attacks, and others have a low resistance to large-scale attacks. Besides these issues, most of these algorithms rely on traditional moment computation, which suffers from numerical error accumulation, leading to numerical instabilities, and time consumption and affecting the performance of these moment-based zero-watermarking techniques. In this paper, we derived multi-channel Gaussian–Hermite moments of fractional-order (MFrGHMs) to solve the problems. Then we used a kernel-based method for the highly accurate computation of MFrGHMs to solve the computation issue. Then, we constructed image features that are accurate and robust. Finally, we presented a new zero-watermarking scheme for color medical images using accurate MFrGHMs and 1D Chebyshev chaotic features to achieve lossless copyright protection of the color medical images. We performed experiments where their outcomes ensure the robustness of the proposed zero-watermarking algorithms against various attacks. The proposed zero-watermarking algorithm achieves a good balance between robustness and imperceptibility. Compared with similar existing algorithms, the proposed algorithm has superior robustness, security, and time computation.

## 1. Introduction

Multimedia tools, imaging technology, and digital image processing are widely used in modern smart healthcare systems. As a result, many medical images and data are transmitted between locations via healthcare networks and the internet. Medical images are used to help clinicians in diagnosis. Most of these medical images contain personal privacy details, which may be maliciously intercepted and tampered with by illegal elements during transmission, leading to patient privacy leakage [1,2]. Digital watermarking technology represents a reliable tool to overcome this challenge [3]. Digital watermarking technology addresses the essential requirements of copyright protection, authentication, integrity verification, and unique identification of medical images during transmission over insecure networks and storage in very large distributed databases. In the medical field, digital watermarking algorithms were applied to protect patients’ privacy, prevent tampering with medical data, and ensure the copyright of medical images [4,5,6,7]. Traditional embedding watermarking algorithms [8,9,10,11] embed digital information that identifies the ownership into the host image/carrier and then ensures the copyright or protects the copyright through extracting the embedded information. Extraction of the embedded information could destroy the information integrity of the host image, especially the fine data, which can lead to an inaccurate diagnosis.

Moreover, for transparency, the amount of embedded information is limited. The host images are distorted in those abovementioned traditional watermarking algorithms. In specific applications such as medical computer-aided diagnosis systems, scanning of artwork, and the military, distortion of images is unacceptable. Therefore, these traditional algorithms are less efficient and not suitable for copyright protection of medical images [12].

For medical image watermarking algorithms, imperceptibility and robustness are particularly important. Consequently, lossless copyright protection is imperative. Wen et al. [13] presented a zero-watermarking (lossless watermarking) concept to overcome the integrity challenge in conventional watermarking algorithms. In this approach, the original image features are extracted without any modifications even after attacks, which shows its reliability for the authentication of medical images. The zero-watermarks are invisible. In addition, it can address the noticeable conflict between imperceptibility and robustness in the traditional watermarking methods [13,14,15]. The zero-watermarking algorithms for digital images could be applied either in the spatial domain [16] or the frequency domain [17,18,19].

Based on the remarkable characteristics of orthogonal moments and their invariants in extracting the features of a medical digital image, moment-based zero-watermarking algorithms have received much attention from researchers. Here, we review the existing moments-based zero-watermarking algorithms, focusing on techniques. We discuss their effectiveness, limitations, and drawbacks. Wang et al. [20] extended the GOFMMs, defined the QGOFMMs, and then extracted the 4D features of the carrier medical image. The extracted images were applied to the zero-watermarking of medical images.

The authors [21] used invariant QPHTs of the color medical images and presented a robust zero-watermarking algorithm. In [22], they utilized QPHFTs in zero-watermarking of three medical CT images. Again, in [23], they applied a geometrically invariant QPHFMs-based null-watermarking to medical images for copyright protection.

Ma et al. [24] protected two similar medical images using a zero-watermarking algorithm based on TPCET and chaotic mapping. Xia et al. [25] applied multiple zero-watermarking of color medical images using QPHFMs. In [26], Niu et al. designed a fast QLPRHFMs-based zero-watermarking for color medical/non-medical images. 

Recently [27,28,29], scientists have demonstrated that orthogonal moments with fractional orders outperform their corresponding ones with integer orders. The authors in [30] designed multiple zero-watermarking using the MFrGMs. In [31], the authors designed FoRHFMs-based lossless copyright protection for medical grayscale images. In [32], Wang and his co-authors derived QFPJFMs for color images and then applied these moments in a zero-watermarking algorithm.

Despite the considerable progress in the moment-based zero-watermarking approaches, the moment-based image zero-watermarking systems have several major drawbacks that limit the system’s performance. Through analysis of each zero-watermarking technique, we can easily summarize the limitations and drawbacks as follows:(1)Most of the abovementioned moment-based zero-watermarking techniques fail to achieve an acceptable trade-off between imperceptibility and robustness performance.(2)Most of them utilized the inaccurate zeroth-approximation method to compute the continuous orthogonal moments of integer orders, which results in a group of inaccurate features.(3)Most of them show low robustness against geometric and noise attacks.(4)Most of them are limited to the standard gray and color images and are not used in the medical field.(5)Due to the inaccurate direct computation method used in these algorithms, zero-watermarking algorithms are time-consuming and unsuitable for telemedicine applications.

These drawbacks have a major influence on the moments-based watermarking system.

Three relevant and interesting research questions that have been designed are given below:

**RQ1**. What is the influence of using a new accurate image descriptor, MFrGHMs, on the robustness of zero-watermarking?

**RQ2.** What is the influence of MFrGHMs on the imperceptibility of zero-watermarking?

**RQ3.** What is the influence of MFrGHMs on the computation time of zero-watermarking?

Motivated by considering the above limitations, this paper aims to develop a new color medical image zero-watermarking algorithm by integrating a new multi-channel fractional-order orthogonal moment with a chaotic map.

Firstly, we derived a new set of MFrGHMs for color medical images. We then used these new moments in designing a new zero-watermarking algorithm. We utilized a kernel-based for highly accurate computation of the MFrGHMs for the carrier medical image. Then, the carrier image was represented by extracting the robust and accurate features of MFrGHMs. Next, the features were transformed into a robust binary feature matrix.

Meanwhile, a 1D Chebyshev map was used in scrambling the watermark to provide a high level of security. Finally, we applied a bitwise exclusive-or operation to generate the zero-watermark for ownership verification. Based on the performed experiments, the proposed algorithm outperformed the existing zero-watermarking algorithms in terms of robustness against various attacks and execution times.

The overall contributions of this paper can be summarized as follows:We present a new image descriptor called multi-channel Gaussian–Hermite moments of fractional orders (MFrGHMs).We utilize a fast and highly accurate kernel-based method to compute (MFrGHMs).We propose a zero-watermarking scheme via accurate features of MFrGHMs, then apply it to protect the color medical image.We apply a new 1D Chebyshev chaotic map to enhance the security levels of the proposed algorithm.The utilization of multi-channel moments significantly reduces the computational complexity.Results from numerous experiments indicate that the proposed algorithm has superiority in robustness, security, and time computation.

This paper is organized into five sections. We describe the MFrGHMs in the second section. A detailed description of the proposed zero-watermarking algorithms is presented in the third section. Experiments and results are discussed in the fourth section. The conclusion is drawn in the fifth section.

## 2. Gaussian–Hermite Moments

This section briefly describes the conventional Gaussian–Hermite moments of integer and fractional orders for gray images. Then, we present the new multi-channel Gaussian–Hermite moments of fractional order for color images. Finally, we describe a highly accurate method to calculate the MFrGHMs coefficients for color medical images.

### 2.1. Traditional Gaussian-Hermite Moments of Gray Images

Hermite polynomials with integer order are defined over the domain (−∞,∞); the Hermite polynomial of the *p*-th degree is [33]:(1)$$H_{p}\left(x\right)={\left(-1\right)}^{p}e^{x^{2}}\frac{d^{p}}{dx^{p}}(e^{-x^{2}}),\;\;\;\;\;\left(p\geq 0\right)$$

In explicit form,
(2)$$H_{p}\left(x\right)=p!\overset{\left\lfloor \frac{p}{2}\right\rfloor }{\underset{m=0}{\sum }}\;{\left(-1^{\;}\right)}^{m}\frac{1}{m!\left(p-2m\right)!}{\left(2x\right)}^{p-2m}$$

The recurrence relation of Hermite polynomials is:(3)$$H_{p+1}\left(x\right)=2xH_{p}\left(x\right)-2\left(p-1\right)H_{p-1}\left(x\right),\;\mathrm{for}\;p\geq 1,$$
where the first two polynomials are $H_{0}\left(x\right)=1$ and $H_{1}\left(x\right)=2x$. With a Gaussian weight function, $e^{-x^{2}}$, Hermite polynomials are orthogonal as follows:(4)$$\int_{-\infty }^{\infty }\;H_{p}\left(x\right)H_{q}\left(x\right)e^{-x^{2}}dx=2^{p}p!\sqrt{\pi }\delta_{pq}$$
where $\delta_{pq}$ is the Kronecker symbol. With the help of Equation (4), the Gaussian–Hermite Polynomials are:(5)$${\hat{H}}_{p}\left(x\right)=\frac{1}{\sqrt{2^{p}p!\sqrt{\pi }}}e^{\left(-\frac{x^{2}}{2}\right)}H_{p}\left(x\right)$$
where
(6)$$\int_{-\infty }^{\infty }\;{\hat{H}}_{p}\left(x\right){\hat{H}}_{q}\left(x\right)dx=\delta_{pq}\;$$

Replacing x by $\frac{x}{\sigma }$, yields:(7)$${\hat{H}}_{p}\left(\frac{x}{\sigma }\right)=\frac{1}{\sqrt{2^{p}p!\sigma \sqrt{\pi }}}e^{\left(-\frac{x^{2}}{2\sigma^{2}}\right)}H_{p}\left(\frac{x}{\sigma }\right)$$

Hence, the Gaussian–Hermite moment (GHM) $G_{pq}$ of image intensity function, f(x, y) is defined as follows [34]:(8)$$G_{pq}\left(f_{\;}\right)=\int_{-\infty }^{\infty }\;\int_{-\infty }^{\infty }\;f_{\;}\left(x,y\right){\hat{H}}_{p}\left(\frac{x}{\sigma }\right){\hat{H}}_{q}\left(\frac{y}{\sigma }\right)dxdy$$

The approximated Gaussian–Hermite moments, ${\tilde{G}}_{pq}$, a digital image of size M×N is calculated using the formula below.
(9)$${\tilde{G}}_{pq}\left(f_{\;}\right)=\overset{M}{\underset{i=0}{\sum }}\overset{N}{\underset{j=0}{\sum }}\;f_{\;}\left(x_{i},y_{j}\right){\hat{H}}_{p}\left(\frac{x_{i}}{\sigma }\right){\hat{H}}_{q}\left(\frac{y_{j}}{\sigma }\right)\Delta x\Delta y$$

### 2.2. Fractional-Order Gaussian–Hermite Moments of Gray Images

According to the remarkable work [35], the FrGHPs of degree *p* are obtained by replacing $x=2t^{\alpha }-1$ in Equation (7):(10)$$FH_{p}^{\left(\alpha \right)}\left(t;\sigma \right)=\sqrt{2\alpha t^{\alpha -1}}{\hat{H}}_{p}\left(2t^{\alpha }-1;\sigma \right)$$

As proved in [35], FrGHPs are orthogonal in [0,1]:(11)$$\int_{0}^{1}\;FH_{p}^{\left(\alpha \right)}\left(t;\sigma \right)FH_{q}^{\left(\alpha \right)}\left(t;\sigma \right)dt=\delta_{pq}\;$$
where the fractional parameter, $\alpha \epsilon R^{+}$ and t∈[0, 1]. The functions, $FH_{p}^{\left(\alpha \right)}\left(t;\sigma \right)$, defined as in explicit form:(12)$$\begin{array}{l} FH_{p}^{\left(\alpha \right)}\left(t;\sigma \right)\\ =\frac{1}{\sqrt{2^{p}p!\sigma \sqrt{\pi }}}e^{\left(-\frac{{\left(2t^{\alpha }-1\right)}^{2}}{2\sigma^{2}}\right)}p!\overset{\left\lfloor \frac{p}{2}\right\rfloor }{\underset{m=0}{\sum }}\;{\left(-1^{\;}\right)}^{m}\frac{1}{m!\left(p-2m\right)!}{\left(\frac{2t^{\alpha }-1}{\sigma }\right)}^{p-2m} \end{array}$$

The recurrence formula of $FH_{p}^{\left(\alpha \right)}\left(t;\sigma \right)$ is:(13)$$FH_{p+1}^{\left(\alpha \right)}{}_{\;}\left(t;\sigma \right)=2\left(2t^{\alpha }-1\right)FH_{p-1}^{\left(\alpha \right)}\left(t;\sigma \right)-2\left(p-1\right)FH_{p-1}^{\left(\alpha \right)}{}_{\;}\left(t;\sigma \right),\;\mathrm{for}\;p\geq 1$$

For p≥1.

With:(14)$$FH_{0}^{\left(\alpha \right)}\left(t;\sigma \right)=\frac{1}{\sqrt{\sigma \sqrt{\pi }}}e^{\left(-\frac{{\left(2t^{\alpha }-1\right)}^{2}}{2\sigma^{2}}\right)}\sqrt{2\alpha t^{\alpha -1}},\;FH_{1}^{\left(\alpha \right)}\left(t;\sigma \right)=2\frac{1}{\sqrt{2^{\;}\sigma \sqrt{\pi }}}e^{\left(-\frac{{\left(2t^{\alpha }-1\right)}^{2}}{2\sigma^{2}}\right)}\sqrt{2\alpha t^{\alpha -1}}\left(2t^{\alpha }-1\right).$$

Based on FrGHPs, the Fractional-order Gaussian–Hermite moment (FrGHM) of image intensity function, f(x, y), is defined as:(15)$$FG_{pq}\left(f_{\;}\right)=\int_{0}^{1}\;\int_{0}^{1}\;f_{\;}\left(x,y\right)FH_{p}^{\left(\alpha \right)}\left(x;\sigma \right)FH_{q}^{\left(\alpha \right)}\left(y;\sigma \right)dxdy$$
where $f_{\;}\left(x,y\right)$ is defined in [0, 1]×[0, 1].

### 2.3. Accurate Computation of New Multi-Channel Gaussian-Hermite Moments of Color Images 

Representing the input color images,$\;g_{C}\left(x,y\right)$ by using the approaches of multi-channel [36,37], where C∈{R,G,B} and $g_{C}\left(x,y\right)=\left\{g_{R}\left(x,y\right),\;g_{G}\left(x,y\right),g_{B}\left(x,y\right)\right\}$. We generalized FrGHM of image intensity function, f(x,y), in Equation (15) to be suitable for the RGB color image, $g_{C}\left(x,y\right)$ and produced the MFrGHMs for $g_{C}\left(x,y\right)$, which are defined as:(16)$$MFG_{pq}\left(g_{C}\right)=\int_{0}^{1}\;\int_{0}^{1}\;g_{C}{}_{\;}\left(x,y\right)FH_{p}^{\left(\alpha \right)}\left(x;\sigma \right)FH_{q}^{\left(\alpha \right)}\left(y;\sigma \right)dxdy$$
where p, q, refer to non-negative integers’ indices.

Accurate computation of the MFrGHMs is the most effective process in a robust zero-watermarking algorithm. An accurate method [38] is used to compute MFrGHMs to achieve this goal. For the original color image, $g_{C}\left(x,y\right)$, we defined a discrete image $g_{C}\left(i,j\right)$ in a domain $\left(x_{i}\;,\;y_{j}\right)\in \;{\left[0,\;1\right]}^{2}$, where i,j=1, 2, 3,…, N. Therefore, the points $\left(x_{i}\;,\;y_{j}\right)$ defined as:(17)$$x_{i}=\frac{i}{N}+\frac{\Delta x}{2}$$
(18)$$y_{j}=\frac{j}{N}+\frac{\Delta y}{2}$$

With $\Delta x=\frac{1}{N}$ and $\Delta y=\frac{1}{N}$.

Equation (16) can be reformulated:(19)$$MFG_{pq}\left(g_{C}\right)=\overset{N}{\underset{i=1}{\sum }}\;\overset{N}{\underset{j=1}{\sum }}\;\;T_{pq}\left(x_{i},y_{j}\right)g_{C}\left(x_{i},y_{j}\right)$$
where:(20)$$T_{pq}\left(x_{i},y_{j}\right)=\int_{x_{i}-\frac{\Delta x}{2}}^{x_{i}+\frac{\Delta x}{2}}\;\int_{y_{j}-\frac{\Delta y}{2}}^{y_{j}+\frac{\Delta y}{2}}\;FH_{p}^{\left(\alpha \right)}\left(x;\sigma \right)FH_{q}^{\left(\alpha \right)}\left(y;\sigma \right)dxdy$$

Because the double integral in Equation (20) is separable; Equation (19) can be written as follows:(21)$$MFG_{pq}\left(g_{C}\right)=\overset{N}{\underset{i=1}{\sum }}\;\overset{N}{\underset{j=1}{\sum }}\;\;IX_{p}\left(x_{i}\right)IY_{q}\left(y_{j}\right)g_{C}\left(x_{i},y_{j}\right)$$
where:(22)$$IX_{p}\left(x_{i}\right)=\int_{x_{i}-\frac{\Delta x}{2}}^{x_{i}+\frac{\Delta x}{2}}\;FH_{p}^{\left(\alpha \right)}\left(x;\sigma \right)dx$$
(23)$$IY_{q}\left(y_{j}\right)=\int_{y_{j}-\frac{\Delta y}{2}}^{y_{j}+\frac{\Delta y}{2}}\;FH_{q}^{\left(\alpha \right)}\left(y;\sigma \right)dy$$

The definite integrals’ limits are:(24)$$U_{i+1}=x_{i}+\frac{\Delta x}{2},\;\;U_{i}=x_{i}-\frac{\Delta x}{2}$$
(25)$$V_{j+1}=y_{j}+\frac{\Delta y}{2},\;V_{j}=y_{j}-\frac{\Delta y}{2}$$

By substitutions of the Equations (24) and (25) in (22) and (23), we get:(26)$$IX_{p}\left(x_{i}\right)=\int_{U_{i}}^{U_{i+1}}\;FH_{p}^{\left(\alpha \right)}\left(x;\sigma \right)\;dx$$
(27)$$IY_{q}\left(y_{j}\right)=\int_{V_{j}}^{V_{j+1}}\;FH_{q}^{\left(\alpha \right)}\left(y;\sigma \right)dy$$

Due to the impossibility of solving the integration in Equations (26) and (27) analytically, we use the accurate Gaussian quadrature method to compute the kernels, $IX_{p}\left(x_{i}\right)$ and $IY_{q}\left(y_{j}\right)$. The definite integral, $\int_{a}^{b}\;h\left(z\right)dz$, could be computed as:(28)$$\int_{a}^{b}\;h\left(z\right)dz\approx \frac{\left(b-a\right)}{2}\overset{c-1}{\underset{l=0}{\sum }}\;w_{l}h\left(\frac{a+b}{2},\frac{b-a}{2}t_{l}\right)$$

Substituting Equations (28) into (26) yields:(29)$$\begin{array}{cc} IX_{p}\left(x_{i}\right)=\int_{U_{i}}^{U_{i+1}\;} & FH_{p}^{\left(\alpha \right)}\left(x;\sigma \right)\;dx\\ & \approx \frac{\left(U_{i+1}-U_{i}\right)}{2}\overset{c-1}{\underset{l=0}{\sum }}\;w_{l}FH_{p}^{\left(\alpha \right)}\left(\frac{U_{i+1}+U_{i}}{2}+\frac{U_{i+1}-U_{i}}{2}t_{l}\right) \end{array}$$

Similarly:(30)$$\begin{array}{cc} IY_{q}\left(y_{j}\right)=\int_{V_{j}}^{V_{j+1}}\; & FH_{q}^{\left(\alpha \right)}\left(y;\sigma \right)dy\\ & \approx \frac{\left(V_{j+1}-V_{j}\right)}{2}\overset{c-1}{\underset{l=0}{\sum }}\;w_{l}FH_{q}^{\left(\alpha \right)}\left(\frac{V_{j+1}+V_{j}}{2}+\frac{V_{j+1}-V_{j}}{2}t_{l}\right) \end{array}$$

As we can see from Equations (29) and (30), the calculation of $IX_{p}\left(x_{i}\right)$ & $IY_{q}\left(y_{j}\right)$ is mainly based on $FH_{p}^{\left(\alpha \right)}\left(x;\sigma \right)$ and $FH_{q}^{\left(\alpha \right)}\left(y;\sigma \right)$, respectively. Therefore, we use the recurrence form of fractional-order Gaussian-Hermit polynomials to effectively, accurately, and quickly compute $IX_{p}\left(x_{i}\right)$ & $IY_{q}\left(y_{j}\right)$. Through overcoming the limitations of the explicit form of fractional-order Gaussian-Hermit polynomials, containing many factorial terms that produce numerical errors and instability.

## 3. Proposed Zero-Watermarking Algorithm

The proposed MFrGHMs-based zero-watermarking algorithm aimed to protect the copyright of color medical images. The proposed algorithm is divided into zero-watermarking generation and verification phases, as illustrated in Figure 1 and Figure 2, respectively.

### 3.1. Watermark Generation

Supposing the carrier’s medical image is $g_{C}$, the size is N×N, and the binary watermark image is *W*, W={w(i,j)∈{0,1},0≤i<P, 0≤j<Q}, and the size is P×Q. The specific steps of the zero-watermark generation phase are:**Step 1: Computing the coefficients of MFrGHMs.**

We compute the maximum order $n_{max}$ MFrGHMs of the host image using Equation (19). We have a totally $L={\left(n_{max}+1\right)}^{2}$ features, where a *L*- length vector of the amplitudes is created.


**Step2: Selected Features**


Based on the standard selection criteria in [39] and by employing a secret key, $K_{1}{}_{\;}$, P×Q coefficient, which is the watermark image size, the precise and accurate MFrGHMs coefficient set, S should be $S=\left\{\left|MFG_{pq}\right|,q\neq \;4m,\;m\in \;Z\right\}$. 


**Step 3: Construction of feature vector**


The P×Q of MFrGHMs coefficients generated in **Step 2** is used for the feature image construction. A P×Q number MFrGHMs moments is randomly selected from the calculated MFrGHMs moments (set S), where we calculate the amplitudes and produce the vector A={a(i), 0≤i<P×Q}.


**Step 4: Binary Feature Vector**


The binary feature sequence B={b(i), 0≤i<P×Q} is obtained from the feature vector/sequence A, where:(31)$$B_{i}=\{\begin{array}{l} 1,\;\;\;\;\;\;if\;\;A_{i}\geq T_{\;}\;\;\;\;\;\;\;\;\;\;\;\\ 0,\;\;\;\;\;\;if\;\;\;A_{i}<T_{\;}\;\;\;\;\;\;\; \end{array},\;\;i=1,2,\ldots \ldots \ldots P\times Q\;\;$$
where *T* is the threshold based on the mean value of *A*.

The binarized vector/sequence B is rearranged into a two-dimensional binary feature image, LF, with the size of P×Q.


**Step 5: Watermark Scrambling**


We scrambled the watermark image W using a 1D Chebyshev map to increase the security and broke the pixels-spatial relationship in the watermark image. The outstanding wide range of control parameters and pseudo-randomness has made the Chebyshev map suitable for image scrambling.

Therefore, the scrambling parameters of the 1D Chebyshev map are used to create the security key.

We transformed the watermark image, W, into a one-dimensional sequence, $OW_{\;}=\left\{ow_{\;}{\left(i\right)}_{\;}:0\leq i<P\times Q\right\}$. Next, a 1D Chebyshev map with controlling parameter r and initial value $x_{0\;}$ (key $K_{1}$) used in generating the chaotic sequence, $C=\left\{c{\left(i\right)}_{\;}:0\leq i<P\times Q\right\}$. The chaotic Chebyshev map is defined as:(32)$$x_{n+1}=\cos\left(r\cos^{-1}x_{n}\right),\;x_{n}\in \left[-1,\;1\right]$$
where the control parameter r ≥ 2.

A chaotic sequence, C is binarized to obtain a chaotic binary sequence, $\hat{C}$ as follows:(33)$$\hat{C}\left(i\right)=\{\begin{array}{l} 1,\;\;\;\;\;\;if\;\;\hat{c}\left(i\right)\geq T_{\;}\;\;\;\;\;\;\;\;\;\;\;\\ 0,\;\;\;\;\;if\;\;\;\hat{c}\left(i\right)<T_{\;}\;\;\;\;\;\;\;\;\; \end{array},\left(0\leq i<P\times Q\right)$$
where T is the threshold.

Then, we performed an XOR operation between OW and $\hat{C}$ to get the chaotic scrambled watermark sequence, $O\hat{W}$. Finally, $O\hat{W}$ is converted to the 2D matrix to generate the scrambled watermark ${\hat{W}}_{\;}=\left\{{\hat{w}}_{\;}\left(i,j\right)\in \left\{0,1\right\},0\leq i<P,\;0\leq j<Q\right\}.$


**Step 6: Generation of the Watermark Image**


The performed XOR operation is used in scrambling the watermark image, $\hat{W}$ and the image’s feature, LF to generate the zero-watermark, $W_{zero}$: $W_{zero}=LF⊕{\hat{W}}_{\;}.$

### 3.2. Detection of the Watermark 

We mainly detect the image’s watermark information in the detection phase, independent of the original images.


**Step 1: Computation of the coefficients of MFrGHMs**


We compute the maximum order $n_{max}$, of MFrGHMs for the verified original image,$\;g_{c}^{*}$ using Equation (19).


**Step 2: Selection of the precise coefficients**


Similar to Step 2 in the Zero-watermark generation, the precise and accurate MFrGHMs coefficient set, $S^{*}$ are obtained.


**Step 3: Construction of feature vector**


A P×Q number MFrGHMs coefficients are randomly selected from accurate MFrGHMs coefficients set $S^{*}$ and their amplitudes are calculated, producing feature sequences $A^{*}=\left\{a^{*}\left(i\right),\;0\leq i<P\times Q\right\}$.


**Step 4: Binary Feature Vector**


We binarized the feature vector/sequence $A^{*}$ using Equation (31) to produce $B^{*}=\left\{b^{*}\left(i\right),\;0\leq i<P\times Q\right\}$. Then, the binarized feature vector/sequence $B^{*}$ is transformed into the 2D binary feature image, $LF^{*}$ of the size P×Q.


**Step 5: Generation of the scrambled watermark image**


An XOR operation is utilized on the zero-watermark, $W_{zero}$, and the image’s features, $LF^{*}$ to generate the scrambled version of the retrieved watermark,$\;W^{*}$: $W^{*}=LF^{*}⊕W_{zero}{}_{\;}$.


**Step 6: Extraction of watermark image.**


The reverse process of scrambling can extract and visually check out the watermark. $W^{*}$ is reversely scrambled using the key, $K_{2}$ of 1D-Chebyshev map to obtain the retrieved watermark, $W^{*}$ denoted as $W^{*}=\left\{w^{*}\left(i,j\right)\in \left\{0,1\right\},0\leq i<P,\;0\leq j<Q\right\}$.

## 4. Experimental Results

Experiments were conducted to evaluate the proposed algorithm. We selected 18 color MRI medical images (256 × 256) from the ‘The Whole-Brain Atlas’ [40], as displayed in Figure 3. Watermarks are binary images (32 × 32), shown in Figure 4, where the fractional parameter is set to t = 1.9.

### 4.1. Evaluation Metrics

The PSNR is used to evaluate the distortion of attacked images, where lower PSNR values reflect a high degree of distortion and vice versa. The PSNR is defined as [41]:(34)$$PSNR\left(\;g_{c}^{\;},\;g_{c}^{w}\right)=10{log}_{10}\frac{255^{2}}{MSE}$$
where:(35)$$MSE=\frac{1}{M^{2}}\left(\overset{M}{\underset{\iota =1}{\sum }}\overset{M}{\underset{\kappa =1}{\sum }}{\left[\;g_{c}^{w}\left(\iota ,\kappa \right)-g_{c}^{\;}\left(\iota ,\kappa \right)\right]}^{2}\right)$$
where $g_{c}^{w}$ and $g_{c}^{\;}$ refer to attacked and original images, respectively.

The robustness of the algorithm can be determined by comparing the BER and the NC of the extracted watermark image $w^{*}$, with that of the original image w. BER and NC are defined as:(36)$$\mathrm{BER}=\frac{1}{P\times Q}\left(\overset{P}{\underset{i=1}{\sum }}\;\overset{Q}{\underset{j=1}{\sum }}\;{\left[w\left(i,j\right)-w^{*}\left(i,j\right)\right]}^{2}\right)$$
(37)$$\mathrm{NC}=\frac{\sum_{i=1}^{P}\sum_{j=1}^{Q}\left[w\left(i,j\right)\times w^{*}\left(i,j\right)\right]}{\sum_{i=1}^{P}\sum_{j=1}^{Q}\left[w\left(i,j\right)\right]{\;}^{2}}$$

The optimum values of BER and NC values are 0 and 1, respectively.

### 4.2. Robustness

Three experiments were conducted to test the robustness of the proposed scheme. In the two experiments, we used the color medical image, “ IMAGE1”, of size 256 × 256, as a carrier image and the binary image, “TREE”, of size 32 × 32, as a watermark. For the third experiment, we selected 18 color medical images of size 256 × 256 as carrier images and the binary image, “horse”, of size 32 × 32 as a watermark. In this experiment, we tested the robustness of the proposed zero-watermarking algorithms against various attacks.

#### 4.2.1. Common Signal Processing Attacks

Noise, filtering, JPEG compression, etc., are common signal processing attacks. These attacks change the energy of the image pixels. We performed a few experiments. First, we applied the JPEG compression with various compression ratios, 10, 30, 50, 70 and 90. After the JPEG-compression attack, the attacked images and their corresponding PSNR are shown in Figure 5. Second, we contaminated color medical images with Gaussian noise with variances of 0.005 and 0.02, and also, the “salt and pepper” noise with density 0.005 and 0.02. The noisy images with their PSNR are shown in Figure 6. Finally, the selected color medical images were subjected to different filtering using a small window (3 × 3); median-filtering, Gaussian-filtering, and average-filtering. The filtered images with their PSNR are shown in Figure 7. For simplicity, we summarize these attacks in Table 1.

The proposed algorithm shows a high degree of robustness to various attacks. The retrieved binary watermarks and the calculated values of BER and NC are summarized in Table 2. Table 2 shows that almost all values of BER and NC approach the optimum values, and the rest of the values of BER and NC are equal to 0 and 1, respectively. The retrieved watermarks using the proposed algorithm are very close to the original due to the high accuracy and stability of the MFrGHMs amplitude against various attacks. Therefore, this algorithm resists the different image processing attacks.

#### 4.2.2. Geometric Attacks

Here, the robustness of the proposed zero-watermarking algorithm is evaluated against common geometric attacks. First, we rotated the medical test image by 5°, 15°, 25°, and 45° (Figure 8). Next, we scaled the color medical image with a scaling factor of 0.5, 0.75, 1.5, and 2 (Figure 9). Then, we applied length–width ratio (LWR) changes to the tested images with factors of (1.0, 0.75) and (0.5, 1.0) (Figure 10). Finally, the medical test image was flipped vertically and horizontally (Figure 11). We summarize the common geometric attacks and their parameters in Table 3 for simplicity.

The results concerning the images subjected to geometric attacks are shown in Table 4. Table 4 shows that the retrieved watermarks from the attacked images are almost identical to the original watermarks. The values of BER and NC have approached or are equal to the optimum values of BER and NC, “0 and 1”, respectively. These results ensure the robustness of the proposed algorithm to the geometric attacks.

### 4.3. Robustness Comparison with Similar Algorithms

We computed the average values of BER of 18 images shown in Figure 3. Then compared the obtained results with the latest zero-watermarking algorithms [20,22,23,26,30,32], where the comparison results are reported in Table 5. We notice that the proposed algorithm achieved the lowest BER and exhibits higher-robustness than the existing zero-watermarking algorithms [20,22,23,26,30,32].

### 4.4. Computational Time

The proposed scheme mainly consists of two phases. The first is the generation, while the second is the verification. We calculate the execution time for the 18 carrier images (512 × 512) and the watermarks (32 × 32) using the proposed and the existing methods [20,22,23,26,30,32]. The average computation times are reported in Table 6.

The proposed zero watermark algorithm requires much less time than the existing algorithms [20,22,23,26,30,32].

From all the numerical experiments, the proposed algorithm achieved the best values for the BER, the NC, and the computation time compared with the existing algorithms. The proposed algorithm has advantages over the existing algorithms due to the new descriptor with intrinsic features, the fast and accurate computation method of MFrGHMs, and the effective 1D Chebyshev chaotic map combined in the proposed algorithm.

## 5. Conclusions

In this paper, we have proposed a robust medical color image zero-watermarking algorithm using a new descriptor of medical images and a chaotic map to answer the research questions, RQ1, RQ2, and RQ3, through different numerical experiments. We derived a new set of multi-channel Gaussian–Hermite moments of fractional orders. These new descriptors are used to design an MFrGHMs-based watermarking algorithm for color medical images. The fractional-order orthogonal descriptors enable the extraction of coarse and fine features from the medical images. We used a highly accurate and numerically stable method to compute the MFrGHMs, minimizing common numerical errors. A 1D Chebyshev chaotic map is used to scramble the watermark and break the correlation between the adjacent pixels, improving the watermark’s security. Using multi-channel moments significantly reduces the computational complexity and speeds up the proposed algorithm. Experimental results ensure the robustness and resistance to various kinds of attacks. Furthermore, compared to several existing algorithms, the proposed algorithm achieves a good trade-off between imperceptibility and robustness and has certain superiority in terms of robustness, time complexity, security, and capacity.

For future works, there are some research studies planned. Firstly, the proposed method will be extended to deal with stereoscopic medical images. Secondly, we will utilize a computational method using a fast Fourier transform to improve the computation time and the accuracy of the feature extraction method of medical images. Then, we will improve the security levels by using a new chaotic map. Finally, we will extend the proposed to other real-life applications such as IoT medical systems.

## Figures and Tables

**Figure 1 sensors-22-05612-f001:**
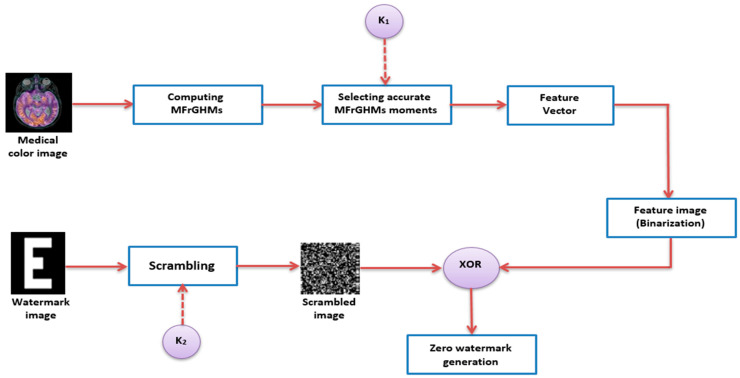
Generation of zero-watermark.

**Figure 2 sensors-22-05612-f002:**
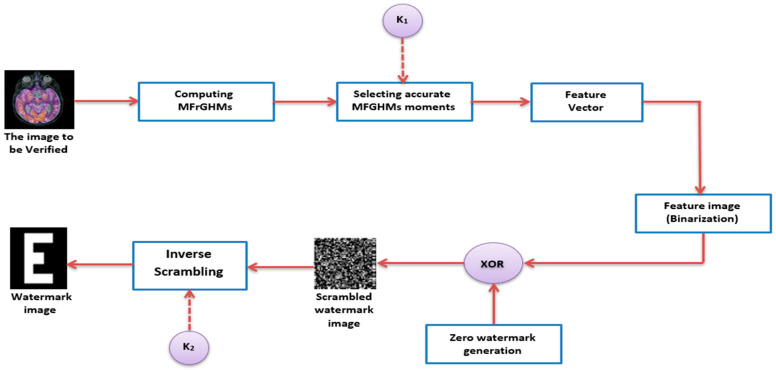
Zero-watermarking verification framework.

**Figure 3 sensors-22-05612-f003:**
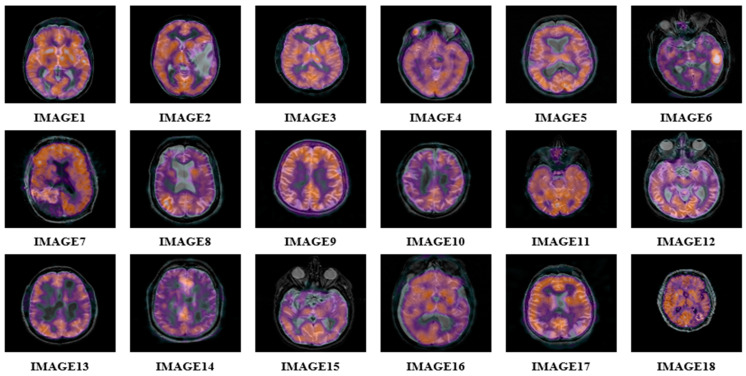
Original MRI color medical images.

**Figure 4 sensors-22-05612-f004:**
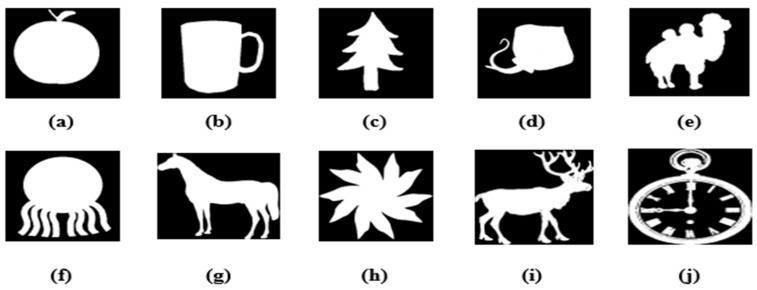
Binary watermarks images.

**Figure 5 sensors-22-05612-f005:**
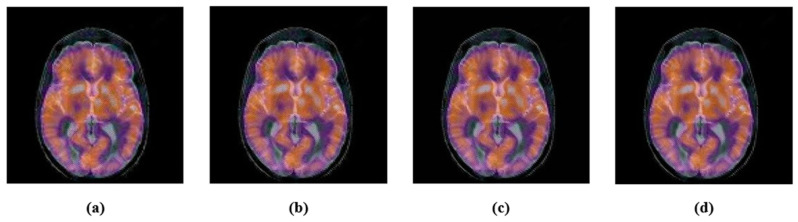
Image after JPEG compression attack: (**a**) JPEG 30, PSNR = 32.1397; (**b**) JPEG 50; PSNR = 34.1217; (**c**) JPEG 70; PSNR = 36.4392; (**d**) JPEG 90; PSNR = 46.7826.

**Figure 6 sensors-22-05612-f006:**
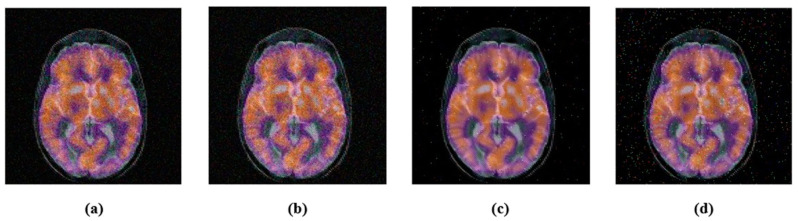
Image after adding noise attack: (**a**) Gaussian-noise 0.005, PSNR = 21.4300; (**b**) Gaussian-noise 0.02, PSNR = 20.8392; (**c**) salt and pepper noise 0.005, PSNR = 27.0980; (**d**) salt and pepper noise 0.02, PSNR = 22.9251.

**Figure 7 sensors-22-05612-f007:**
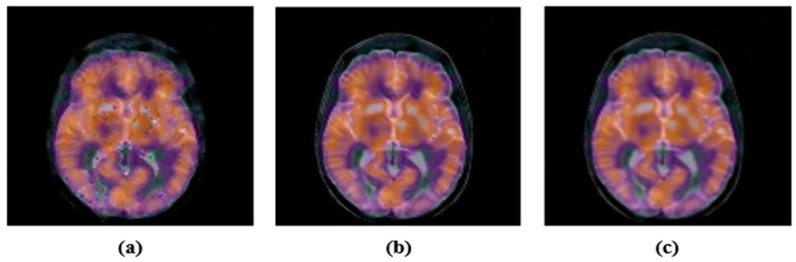
Images after filtering attack: (**a**) median-filtering 3 × 3; PSNR = 24.2301; (**b**) Gaussian-filtering 3 × 3; PSNR = 18.7036; (**c**) average-filtering 3 × 3 PSNR = 19.2364.

**Figure 8 sensors-22-05612-f008:**
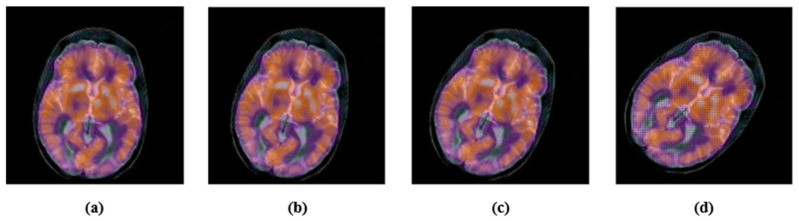
Rotated Images: (**a**) 5°; PSNR = 16.5233; (**b**) 15°; PSNR = 15.1828; (**c**) 25°; PSNR = 14.3906; (**d**) 45°; PSNR = 13.6059.

**Figure 9 sensors-22-05612-f009:**
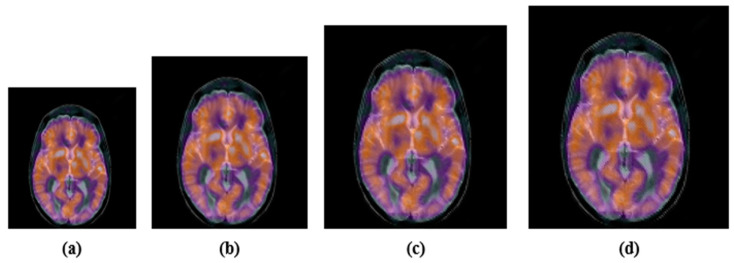
Scaled images: (**a**) 0.5, (**b**) 0.75, (**c**) 1. 5, (**d**) 2.0.

**Figure 10 sensors-22-05612-f010:**
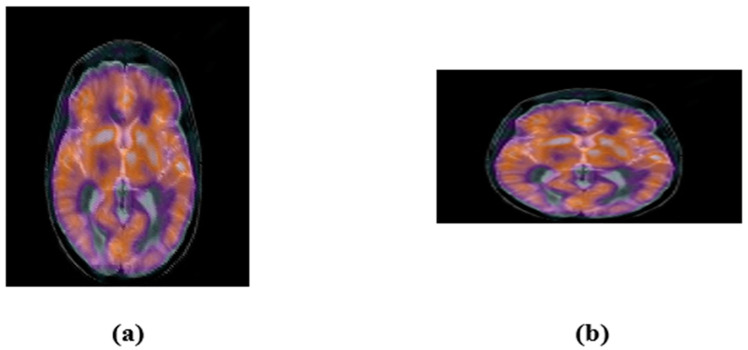
Image after under LWR changing: (**a**) (1.0, 0.75), (**b**) (0.5, 1.0).

**Figure 11 sensors-22-05612-f011:**
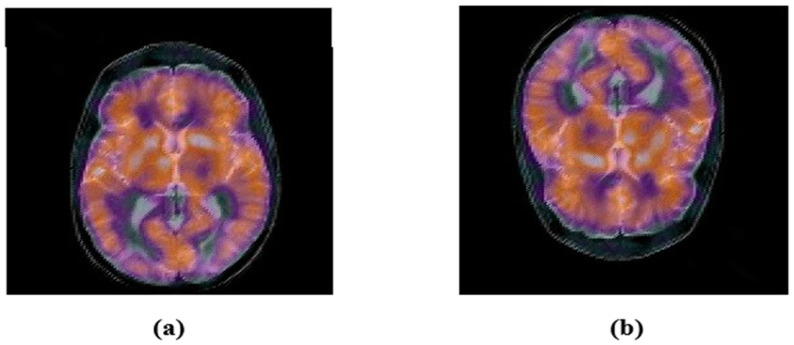
Image after flipping attacks: (**a**) Horizontal-flipping, PSNR = 8.9862, (**b**) Vertical-flipping, PSNR = 12.5725.

**Table 1 sensors-22-05612-t001:** Common image processing attacks and their parameterization.

Attack	Parameter
JPEG-Compression	Compression-Ratio: 30%, 50%, 70%, 90%
Gaussian-noise	Noise-variance: 0.005, 0.02
“Salt and pepper”-noise	Noise-density: 0.005, 0.02
Median-filtering	Window-size: 3 × 3
Gaussian-filtering	Window-size: 3 × 3
Average-filtering	Window-size: 3 × 3

**Table 2 sensors-22-05612-t002:** The extracted watermarks and the corresponding BER and NC values under common image processing attacks.

**Applied-Attack**	**JPEG-(10)**	**JPEG-(30)**	**JPEG-(10)**	**JPEG-(70)**
Extracted watermark	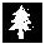	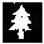	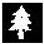	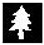
BER	0.0068	0.0039	0.0029	0
NC	0.9849	0.9914	0.9936	1.0000
**Applied-Attack**	**JPEG-(90)**	**“Gaussian-** **Noise”** **(0.005)**	**“Gaussian-** **Noise”** **(0.02)**	**“Salt and pepper”** **noise (0.005)**
Extracted watermark	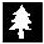	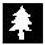	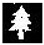	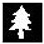
BER	0	0	0.0039	0
NC	1.0000	1.0000	0.9914	1.0000
**Applied-Attack**	**“Salt and pepper”** **noise (0.02)**	**“Median-Filtering”** **(3 × 3)**	**“Gaussian-filtering”** **(3 × 3)**	**“Average-Filtering”** **(3 × 3)**
Extracted watermark	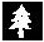	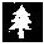	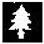	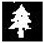
BER	0.0020	0.0029	0.0029	0.0039
NC	0.9957	0.9936	0.9936	0.9914

**Table 3 sensors-22-05612-t003:** Geometric attacks and their parameterization.

Attack	Parameter
Image rotation	Rotation angles: 50, 150, 250, 450
Image scaling	Scaling factors: 0.5, 0.75, 1.25, 1.5
Length–width ratio changing	Factor (1.0, 0.75) and (0.5, 1.0). Note: The parameters refer to vertically and horizontally scaling.
Image flipping	Vertical and horizontal

**Table 4 sensors-22-05612-t004:** The watermarks and corresponding BER and NC values were extracted under common geometric attacks.

**Applied** **Attack**	**Rotation# 5°**	**Rotation# 15°**	**Rotation# 25°**	**Rotation# 45°**
Extracted watermark	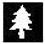	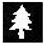	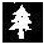	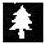
BER	0	0	0.0039	0.0010
NC	1.0000	1.0000	0.9914	0.9979
**Applied** **Attack**	**Scaling# 0.5**	**Scaling# 0.75**	**Scaling# 1.5**	**Scaling# 2.0**
Extracted watermark	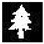	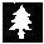	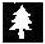	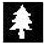
BER	0.0029	0.0020	0.0010	0
NC	0.9936	0.9957	0.9979	1.0000
**Applied** **Attack**	**LWR** **(1.0, 0.75)**	**LWR** **(0.5, 1.0)**	**Horizontal-flipping**	**Vertical-** **Flipping**
Extracted watermark	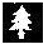	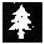	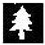	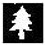
BER	0.0049	0.0068	0.0010	0.0010
NC	0.9892	0.9849	0.9979	0.9979

**Table 5 sensors-22-05612-t005:** Average BER comparison with [20,22,23,26,30,32].

Attacks		Scheme [20]	Scheme [22]	Scheme [23]	Scheme [26]	Scheme [30]	Scheme [32]	Proposed
Rotation	5	0.0000	0.0037	0.0015	0.0000	0.0011	0.0000	0.0000
	45	0.0000	0.0042	0.0044	0.0000	0.0015	0.0000	0.0000
Scaling	0.25	0.0002	0.0274	0.0410	0.0048	0.0054	0.0000	0.0001
	4.0	0.0001	0.0020	0.0001	0.0000	0.0000	0.0000	0.0000
Flipping	H	0.0000	0.0035	0.0027	0.0000	0.0001	0.0000	0.0000
	V	0.0000	0.0048	0.0036	0.0000	0.0002	0.0000	0.0000
Upper left corner cropping	1/16	0.0002	0.0000	0.0000	0.0000	0.0000	0.0000	0.0000
	1/8	0.0050	0.0000	0.0000	0.0000	0.0000	0.0000	0.0000
JPEG compression	30	0.0007	0.0054	0.0079	0.0000	0.0034	0.0000	0.0002
	70	0.0009	0.0037	0.0030	0.0000	0.0020	0.0011	0.0000
Median filtering	3 × 3	0.0039	0.0098	0.0342	0.0048	0.0035	0.0038	0.0013
Gaussian filtering	3 × 3	0.0001	0.0024	0.0015	0.0000	0.0015	0.0000	0.0000
Salt and pepper noise	0.01	0.0009	0.0260	0.0249	0.0048	0.0027	0.0011	0.0005
Gaussian-noise	0.01	0.0077	0.0200	0.0273	0.0048	0.0039	0.0106	0.0029

**Table 6 sensors-22-05612-t006:** Computation time for the proposed and the existing algorithms [20,22,23,26,30,32].

	Scheme [20]	Scheme [22]	Scheme [23]	Scheme [26]	Scheme [30]	Scheme [32]	Proposed
Generation time (s)	40.6334	45.952	33.850	25.731	13.851	16.863	11.769
Verification time (s)	41.7840	46.238	34.137	26.148	15.017	17.208	12.493

## Data Availability

Not applicable.

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
