# Peer review of "Robust Zero-Watermarking of Color Medical Images Using Multi-Channel Gaussian-Hermite Moments and 1D Chebyshev Chaotic Map"

_sensors, 2022, doi:10.3390/s22155612_

Round 1

Reviewer 1 Report

1-The article layout is very poorly done. The layout should be redone and the equations should be written more accurately.

2-The motivation and contribution of the article should be expressed more clearly.

3-What's new other than defining "Gaussian-Hermite Moments" in RGB channels?

4-The algorithm should be written in algorithm format.

5-PSNR should be referenced.

https://www.sciencedirect.com/science/article/pii/S0020025521013475

6-"Table (7): Abbreviations" should be given at the beginning of the intro.

Author Response

Respond to the comments of Reviewers

Dear Editor-in-Chief

Sensors

Thank you, the handling editor, and the anonymous reviewers for their constructive comments, which helped me and my co-author further clarify and enhance the paper's comprehensiveness. We considered their considerations in preparing my answer and revising the manuscript. Original comments of reviewers are presented in conjunction with my responses. All revised and newly added sections are highlighted in red color.

Reviewer#1:

Comment (1):

The article layout is very poorly done. The layout should be redone and the equations should be written more accurately.

Respond (1):

The authors thank reviewer#1 for this valuable comment. We revised and improved the manuscript accordingly.

Comment (2):

The motivation and contribution of the article should be expressed more clearly.

Respond (2):

The authors thank reviewer#1 for this valuable comment. In the revised manuscript, the authors re-wrote the motivation and contribution of this manuscript in a clearer form.

Comment (3):

What's new other than defining "Gaussian-Hermite Moments" in RGB channels?

Respond (3):

The authors would thank reviewer#1 for this valuable comment. In the revised manuscript, the authors added more explanations about the contribution of this manuscript.

The overall contributions of this paper can be summarized as follows:

  • We present a new image descriptor called multi-channel Gaussian-Hermite moments of fractional orders (MFrGHMs.(
  • We utilize a fast and highly accurate kernel-based method to compute (MFrGHMs).
  • We propose a zero-watermarking scheme via accurate features of MFrGHMs, then apply it to protect the color medical image.
  • We apply a new 1-D Chebyshev chaotic map to enhance the security levels of the proposed algorithm.
  • The utilization of multi-channel moments significantly reduces the computational complexity.
  • Results from numerous experiments indicate that the proposed algorithm has superiority in robustness, security, and time computation.

Comment (4):

The algorithm should be written in algorithm format.

Respond (4):

The authors thank reviewer#1 for this valuable comment. We only used generalized steps to discuss in more detail the proposed method and the Figures (1) and (2) summarize the main steps of the proposed.

Comment (5):

PSNR should be referenced.

https://www.sciencedirect.com/science/article/pii/S0020025521013475

Respond (5):

The authors thank reviewer#1 for this valuable comment. We added this remarkable reference [36].

Comment (6):

Table (7): Abbreviations" should be given at the beginning of the intro.

Respond (6):

The authors thank reviewer#1 for this valuable comment. Table (7) is given in the revised manuscript at the beginning of the intro.

Reviewer 2 Report

The authors proposed a Robust Zero-Watermarking of Color Medical Images using Multi-channel Gaussian-Hermit Moments and 1-D Chebyshev Chaotic Map. The work is interesting and can be published subject to minor changes. My observations are:

1. The abstract should need to rewrite. The discussion about results is missing in that.

2. There are some typos that need to fix throughout the manuscript. I suggest using an English language proofreader to avoid ambiguity in some sentences.

3. Fig 1 needs to elaborate to understand the methodology. The quality of the figures are poor.

4. How the feature vector is obtained is unclear.

5. Is section 4.1 objective function? or performance parameters.

6Literature needs to elaborate by adding some more recent relevant works such as

 "Secure video communication using firefly optimization and visual cryptography"

7. You can add more geometrical attacks to test the performance of your work.

8. Compare the work with above-suggested works.

Author Response

Respond to the comments of Reviewers

Dear Editor-in-Chief

Sensors

Thank you, the handling editor, and the anonymous reviewers for their constructive comments, which helped me and my co-author further clarify and enhance the paper's comprehensiveness. We considered their considerations in preparing my answer and revising the manuscript. Original comments of reviewers are presented in conjunction with my responses. All revised and newly added sections are highlighted in red color.

Reviewer#2:

Comment (1):

The abstract should need to rewrite. The discussion about results is missing in that.

Respond (1):

The authors thank reviewer#2 for this valuable comment. In the revised manuscript, we revised the abstract according to this valuable comment.

Comment (2):

There are some typos that need to fix throughout the manuscript. I suggest using an English language proofreader to avoid ambiguity in some sentences.

Respond (2):

The authors would thank reviewer#2 for this valuable comment. We consulted an English instructor, and the manuscript is revised accordingly.

Comment (3):

Fig 1 needs to elaborate to understand the methodology. The quality of the figures are poor.

Respond (3):

The authors would thank reviewer#2 for this valuable comment. We added an improved version of Fig.1 and the other figures in the revised manuscript.

Comment (4):

How the feature vector is obtained is unclear.

Respond (4):

The authors thank reviewer#2 for this valuable comment. We added an explanation of the feature vector construction in the revised manuscript.

Comment (5):

Is section 4.1 objective function? or performance parameters.

Respond (5):

The authors thank reviewer#2 for this valuable comment. The objective function is suitable for the minimization algorithm, where we replaced the objective evaluation with the most suitable "Evaluation Metrics "in the revised manuscript.

Comment (6):

Literature needs to elaborate by adding some more recent relevant works such as "Secure video communication using firefly optimization and visual cryptography".

Respond (6):

The authors thank reviewer#2 for this valuable comment. In the revised manuscript, we added the proper reference [8].

Comment (7):

You can add more geometrical attacks to test the performance of your work.

Respond (7):

The authors thank reviewer#2 for this valuable comment. In the manuscript, we added more geometrical attacks.

Comment (8):

Compare the work with above-suggested works.

Respond (8):

The authors thank reviewer#2 for this valuable comment. In the revised manuscript, we added the proper reference of the above-suggested work, "Secure video communication using firefly optimization and visual cryptography," in the introduction section to revise the literature. The suggested work applied the embedding technique to secure the standard video. In contrast, the proposed algorithm applied the zero watermarking technique to color medical images.

Reviewer 3 Report

The authors report a Robust Zero-Watermarking of Color Medical Images using Multi-channel Gaussian-Hermit Moments and 1-D Chebyshev Chaotic Map. Moreover, they derived multi-channel Gaussian-Hermit moments of fractional-order (MFrGHMs), and then they used a kernel-based method for the highly accurate computation of MFrGHMs. The manuscript is not written professionally and the aims of the work are not clear. I believe most of this manuscript’s work has been done before and there is no novelty. However, some shortcomings and problems exist as follows:

  1. Subsection 2.1 is not written clearly and NEEDS to be rewritten. For example, defining Eq. (1) needs to be written correctly. The connection between Eqs. (4) and (1) should be addressed. 
  2. Since the manuscript is discussing digital images, it is better to consider Eq. (8) in its digital form not analog.
  3. Subsection 2.3 is too short and not clear. If the authors claim that this proposed approach for color images is novel, they have to describe it well and clearly.
  4. What is the novelty of this work? For theoretical works like this, the authors need to bring some novel and new ideas instead of applying the existing works to some medical images.
  5. I am not sure the authors have used accurate computation of MFrGHMs with help of [32]. They should show some original approaches that [32] used for it.
  6. What is the authors’ contribution to their proposed work in Section 3?
  7. Why this approach is only suitable for medical images? Could use this technique for grayscale images?
  8. The English of the manuscript needs to be improved. 

Author Response

Respond to the comments of Reviewers

Dear Editor-in-Chief

Sensors

Thank you, the handling editor, and the anonymous reviewers for their constructive comments, which helped me and my co-author further clarify and enhance the paper's comprehensiveness. We considered their considerations in preparing my answer and revising the manuscript. Original comments of reviewers are presented in conjunction with my responses. All revised and newly added sections are highlighted in red color.

Reviewer#3:

Comment (1):

Subsection 2.1 is not written clearly and NEEDS to be rewritten. For example, defining Eq. (1) needs to be written correctly. The connection between Eqs. (4) and (1) should be addressed.

Respond (1):

The authors thank reviewer#3 for this valuable comment. In the revised manuscript, we re-wrote the equations correctly and revised Subsection 2.1.

Equation (1) is a Rodrigues representation of Hp(x) and represents the Hermite polynomial based on the differential equation. In particular, Equations (2) or (3) are used. Equation (4) gave the orthogonality property of Hermite polynomial. Many mathematical papers prove the orthogonality in Eq.(4). Therefore, in our work or related work, we only need the orthogonality property of Hermite polynomial.

Comment (2):

Since the manuscript is discussing digital images, it is better to consider Eq. (8) in its digital form not analog.

Respond (2):

The authors thank reviewer#3 for this valuable comment. Equation (8) represents the exact form of Gaussian–Hermite moments in the continuous domain. In the revised manuscript, the authors added equation (9) to represent equation (8) for a digital image in the discrete domain.

Comment (3):

Subsection 2.3 is too short and not clear. If the authors claim that this proposed approach for color images is novel, they have to describe it well and clearly.

Respond (3):

The authors would thank reviewer#3 for this valuable comment. Subsection 2.3 is too short because equation (16) is the extension of equation (15) in the revised manuscript. Equation (15) is used for the grayscale image, while equation(16) is used for three channels, the RGB image. In the revised manuscript, we added an explanation about it.

Comment (4):

What is the novelty of this work? For theoretical works like this, the authors need to bring some novel and new ideas instead of applying the existing works to some medical images.

Respond (4):

The authors thank reviewer#3 for this valuable comment. Besides this, In the revised manuscript, we added more explanations about the contribution of this manuscript. The main objective of our work is to design an accurate and robust applicable watermarking scheme for a real-life application such as watermarking medical color images.

The overall contributions of this paper can be summarized as follows:

  • We present a new image descriptor called multi-channel Gaussian-Hermite moments of fractional orders (MFrGHMs.(
  • We utilize a fast and highly accurate kernel-based method to compute (MFrGHMs).
  • We propose a zero-watermarking scheme via accurate features of MFrGHMs, then apply it to protect the color medical image.
  • We apply a new 1-D Chebyshev chaotic map to enhance the security levels of the proposed algorithm.
  • The utilization of multi-channel moments significantly reduces the computational complexity.
  • Results from numerous experiments indicate that the proposed algorithm has superiority in robustness, security, and time computation.

Comment (5):

I am not sure the authors have used accurate computation of MFrGHMs with help of [32]. They should show some original approaches that [32] used for it.

Respond (5):

The authors thank reviewer#3 for this valuable comment. In the revised manuscript, the integration in equations (26) and (27) is impossible to compute exactly. Therefore, we used an accurate numerical integration method as Gaussian quadrature to compute equations (26) and (27).

Comment (6):

What is the authors' contribution to their proposed work in Section 3?.

Respond (6):

The authors thank reviewer#3 for this valuable comment. Section 3 presents an accurate and robust zero watermarking for medical images. The proposed method is based on a new image descriptor called multi-channel Gaussian-Hermite moments of fractional orders (MFrGHMs) to enhance the proposed algorithm's security levels. We used a new 1-D Chebyshev chaotic map.

Comment (7):

Why this approach is only suitable for medical images? Could use this technique for grayscale images?.

Respond (7):

The authors thank reviewer#3 for this valuable comment. Our approach can be used for medical and standard color images. In this work, we applied the proposed medical images because of the importance of this kind of image in our life. And the proposed algorithm preserves the quality of host color images. Therefore, the proposed is suitable for any image, especially color medical images. We can use this technique for grayscale images; in this case, we apply the proposed for a single channel instead of three channels RGB.

Comment (8):

The English of the manuscript needs to be improved.

Respond (8):

The authors thank reviewer#3 for this valuable comment. We consulted an English instructor, and the manuscript is revised accordingly.

Finally, the authors would like to thank the Editor-in-chief,

the handling editor, and the anonymous reviewers

for their time, efforts, and valuable comments.

Round 2

Reviewer 1 Report

Thank you to the authors for the revision. Paper can be published.

Author Response

Thank you for your recommendation for acceptance.

Reviewer 3 Report

Most of comments are improved by the authors. Still some subsections are too short and not necessary for this paper as I mentioned before.

Author Response

Respond to the comments of Reviewers

Dear Editor-in-Chief

Sensors

Thank you, the handling editor, and the anonymous reviewers for their constructive comments, which helped me and my co-author further clarify and enhance the paper's comprehensiveness. We considered their considerations in preparing my answer and revising the manuscript. Original comments of reviewers are presented in conjunction with my responses. All revised and newly added sections are highlighted in red color.

Reviewer#3:

Comment:

Most of comments are improved by the authors. Still some subsections are too short and not necessary for this paper as I mentioned before.

Respond (1):

The authors thank reviewer#3 for this valuable comment. In the revised manuscript, we combined Subsection 2.3 with Subsection 2.4.

Finally, the authors would like to thank the Editor-in-chief,

the handling editor, and the anonymous reviewers

for their time, efforts, and valuable comments.